# Search and Rescue in a Maze-like Environment with Ant and Dijkstra Algorithms

**Zainab Husain** [1], **Amna Al Zaabi** [1], **Hanno Hildmann** [2], **Fabrice Saffre** [3], **Dymitr Ruta** [4] **and A. F. Isakovic** [5,*]

1 Electrical and Computer Eng. Department, Khalifa University (KUST),
  Abu Dhabi P.O. Box 127788, United Arab Emirates
2 Intelligent Autonomous Systems Group, Netherlands Organisation for Applied Scientific Research (TNO),
  2597 AK Den Haag, The Netherlands
3 Technical Research Centre of Finland (VTT), 02150 Espoo, Finland
4 Etisalat—BT Innovation Center (EBTIC), Khalifa University (KUST),
  Abu Dhabi P.O. Box 127788, United Arab Emirates
5 Physics and Astronomy Department, Colgate University, Hamilton, NY 13346, USA
* Correspondence: aisakovic@colgate.edu or iregx137@gmail.com

**Abstract:** With the growing reliability of modern ad hoc networks, it is encouraging to analyze the potential involvement of autonomous ad hoc agents in critical situations where human involvement could be perilous. One such critical scenario is the Search and Rescue effort in the event of a disaster, in which timely discovery and help deployment is of utmost importance. This paper demonstrates the applicability of a bio-inspired technique, namely Ant Algorithms (AA), in optimizing the search time for a route or path to a trapped victim, followed by the application of Dijkstra's algorithm in the rescue phase. The inherent exploratory nature of AA is put to use for faster mapping and coverage of the unknown search space. Four different AA are implemented, with different effects of the pheromone in play. An inverted AA, with repulsive pheromones, was found to be the best fit for this particular application. After considerable exploration, upon discovery of the victim, the autonomous agents further facilitate the rescue process by forming a relay network, using the already deployed resources. Hence, the paper discusses a detailed decision-making model of the swarm, segmented into two primary phases that are responsible for the search and rescue, respectively. Different aspects of the performance of the agent swarm are analyzed as a function of the spatial dimensions, the complexity of the search space, the deployed search group size, and the signal permeability of the obstacles in the area.

**Keywords:** search and rescue; SAR; ant algorithms; ant colony optimization; ACO; maze exploration; UAV; UAS; drones; civil security; public safety; smart city

## 1. Introduction

Drones, often referred to as Unmanned Aerial Systems (UASs) [1], Remotely Piloted Aircrafts (RPAs) [2], Remotely Operated Aircrafts (ROAs) [3] or, most commonly, Unmanned Aerial Vehicles (UAVs) [4], are on their way to becoming pervasive technology. They have been around for quite some time, with their application initially driven by needs in the military domain (the US military used drones as far back as the Vietnam War [5]), where the precise determination of a unit's location is a common challenge in the field [6]. UAVs have recently become commercially available for civilian use, even for hobbyists, and the use of one or more drones for the express purpose of simultaneous localization and mapping (SLAM) [7] is a thriving research area [8].

Nowadays, UAVs are technologically advanced devices capable of autonomous flight operations [9], making *remote* piloting an option, not a requirement. Drones come in all kinds of shapes and forms, as well as vastly different prices, depending on their intended

use and the required specifications [10]. Autonomous UAVs are becoming an increasingly viable alternative to using human labor, especially for data collection purposes, with key application areas being precision agriculture [11], civil defense, such as fire fighting [12], traffic management [13] or to locate victims in the aftermath of a disaster [14].

While many applications can benefit from the use of (semi-)automated aerial devices due to the reduced cost (in comparison to human labor) and ease of access these devices have [15], few applications can make as strong a case as civil defense and disaster response. UAVs are especially useful in the aftermath of a large disaster because they are (a) expendable, (b) fast moving and (c) capable of moving in 3D space. The latter is extremely beneficial when existing infrastructure has been wiped out. Thus, there is increasing attention in the literature regarding the use of drones for, e.g., simultaneous localization and mapping (SLAM) [16], logistic operations [12], recovery of hazardous material [17], traffic management and monitoring, as well as the monitoring of structures and infrastructure and inspecting terrain [18] leading up to the deployment of human operatives into an environment. Furthermore, drones are quite capable of transporting and operating communication infrastructure, making them well suited to the formation of dynamic communication and sensing networks [19,20]. When this is carried out by a swarm, the members of the swarm can use their on-board communication equipment to form temporary dynamic networks, so-called *ad hoc* networks [21,22].

The very nature of a disaster suggests that large, possibly critical damages were caused to the infrastructure. It is not uncommon that the systems needed most in order to react quickly are the worst affected. Natural disasters often result in significant loss of communication and data-collection infrastructure. The use of so-called *swarms* (formations of multiple UAVs that can, to some extent, operate as a single operational unit) [23] is a topic that receives increasing attention. Whether they are acting as mobile sensor networks [13], monitoring personnel or victims [14] or carrying out tracking and surveillance tasks in general [17], the benefits of being able to deploy such systems quickly is evident.

## 2. Background

The use of drones as mobile and airborne sensing platforms in general [9], and with a focus on disaster response and civil defense in specific circumstances [24], has been discussed in the literature [25]. We now provide some background for Search and Rescue (SAR) operations (Section 2.1) and the use of wireless and mobile/ad hoc networks in this context (Section 2.2), and argue that drone-based wireless networks have great potential for this application domain. Finally, we briefly elaborate on the problem of finding the shortest path between certain nodes in a network. In doing so, we argue the benefits of nature-inspired heuristics (Section 2.3).

### 2.1. Indoor Search and Rescue (SAR) Operations for Swarms of Drones

There is a fast-growing body of literature dedicated to various aspects of deploying robots during—or in the aftermath of—a disaster. Until recently, practical considerations ensured that the majority of applications assumed autonomously operating *ground*-based robots. In these 2D environments, solutions to problems such as the collision-free movement of a team of robots [26] in—or through—the environment, or the movement of a team while maintaining some formation [27], are being proposed. This article targets drones (airborne cyber–physical systems) but simplifies the environment to be 2D (we discuss extending this work to 3D in Section 6.2 on future work). We argue that in-flight collision in 3D space could be avoided by simply assigning a different flight altitude to every drone. Since the simulated environment/our model for the environment does not contain vertical obstacles (that can be avoided by moving in the z-axis) this practically allows us to partition the 3D space into horizontal *slices*, providing one for each drone to operate in. The performance results we obtained are therefore expected to translate well into true 3D operations.

The use of UAVs for SAR is proposed in the literature [20] for a number of reasons. These include the handling of materials that are dangerous for humans [17] or located in

environments that are dangerous or inaccessible for humans that need to be inspected first [18]. The flexibility [13] of drones makes them well suited to serving as mobile/aerial urban sensing platforms [23] for SAR operations or for aerial tracking [17], especially in inaccessible environments and for the detection of victims or victims' life signs [14].

### 2.1.1. Swarm (Multi-Agent) SAR Operations

Of course, the use of drones is not restricted to disasters, but as such events often result in partial or total loss of existing communication and data-collection infrastructure and cause substantial damage to buildings, drones have been deployed to provide situational awareness (SA) [18], deliver (medical) supplies [12], and monitor human personnel in the field [14], e.g., fire fighters [12]. For example, the AceCore NEO drone, shown in Figure 1 (center, right) is being used in trials for systems intended for use by the Dutch police forces.

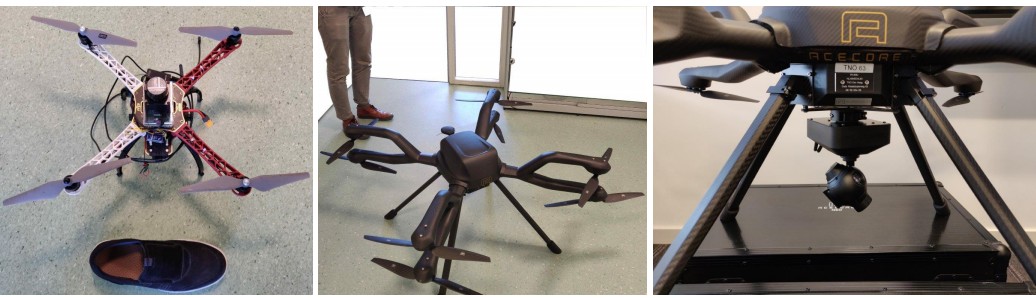

**Figure 1.** Examples of the drones available in the Intelligent Autonomous Systems group at TNO: (**left**) a small device for proof-of-concept testing; shoe for scale. (**Center**) a heavy-duty AceCore NEO drone capable of carrying payloads in excess of 8kg outside and in normal weather conditions. (**Right**) a bottom-mounted gimbal for visual data collection, possibly to be analyzed onboard for fast victim recognition. The presented work in this article has not been deployed under real-world conditions, but TNO and VTT are planning to conduct real-world tests in the near future; cf. [9,24].

More recently, the drop in cost for units, as well as the operation thereof, has made the operation of multiple drones, acting as a single larger collective (often called a *swarm* [23]), a realistic scenario for, e.g., aerial tracking [17]. The authors of [28] investigated SAR operations with ground based robot swarms using actual devices, but while the topic of performing SAR operations using autonomous agents has been an area of widespread research [20], most of this work is still rather general. Due to this, there is still room for improvement in terms of optimizing the search algorithm for specific metrics, such as the speeding up of the rescue. Due to the large potential for unknowns and uncertainties, traditional deterministic approaches are at a disadvantage against heuristics.

Considering temporal and spatial uncertainties and ambiguities in a disaster struck search space (and not the least because of the likelihood of continuing changes in the environment), heuristic techniques, rather than classic, deterministic ones are worth exploring in expectation of a better performance in such unstable environments [29].

### 2.1.2. UAVs in SAR Operations

Especially with regard to disaster response, deployment of assets in the immediate aftermath of an event can be extremely dangerous (for the assets) due to the unstable nature of the situation, as well as simply because the condition thereof is unknown. It has become increasingly common to deploy autonomous technology prior to allowing human personnel into an area, as human intervention might be inefficient or come at a risk of harm to personnel [30]. With this in mind, ref. [31] highlights the ability to sense as an integral basic functionality of drones. The literature makes frequent reference to the potential [32] of using UAVs as autonomous or semi-autonomous operating data acquisition platforms: for example, refs. [33,34] identify a number of different application areas, with SAR operations being among them.

Specifically, the deployment of a UAVs, be it as single units or as a swarm, to act as a node, or to form an entire cell network themselves [35], has been evaluated in the literature. The ability to dynamically adapt the network's topology in reaction to link loss increases resilience against outages and other connectivity problems [36]. With regard to data protection and the security aspects of a network, ref. [37] propose a secure communication protocol for inter-UAV/UAV-sensor communication.

One important application category of such operation are indoor operations, in which assets may have to operate in the absence of—or under the assumption of unreliable or compromised—communication infrastructures [38]. As the safety of the dispatched personnel depends on the ability to keep them apprised of changes in the situation, establishing a reliable means to communicate is often one of the first steps in deployment. When considering the use of drones, a number of challenges have been identified. Timely and efficient creation of infrastructure is often crucial, but hampered by the uncertainty of the condition of the environment. Under the assumption that this is known, establishing the optimal locations for the nodes is relatively straightforward and could be calculated in advance/offline. As this assumption does often not hold, even the collision-free deployment process into an indoor environment [39] (as a prerequisite to surveillance and reconnaissance to establish situational awareness) can be complicated.

### 2.1.3. Path Planning in Maze-like Environments

From an optimization point of view, the search in an ambiguous environment can be modeled as a maze-solving problem where the maze walls denote the presence of unknown obstacles on the way to the rescue target. Path planning through a prior known environment (often considered *"global path planning"*) is a relatively simple and well documented research topic, with several of its main issues having been addressed [40,41]. However, maneuvering a maze with only local information on the search space is called *"local path planning"*, which is performed *on the go* during the maze-solving process. Searching an unknown or uncertain environment is similar to a local path-planning problem, in which heuristic techniques (cf. Section 2.3) can be employed to optimize the search process [42].

### 2.1.4. Communication Routing

Once the maze has been solved (in our scenario and domain, *solving the maze* is equivalent to locating the rescue target) a swarm of ad hoc agents can further accommodate the rescue process by setting up a relay network for the base station to monitor the trapped target. In the literature, communication services have been identified as an integral functionality of swarms [31] and indeed, this can be of significant benefit while a relief team is deployed to reach the victim at the rescue site.

This requires a routing algorithm that can engineer a reverse route between the already dispersed search agents, under relevant communication constraints. There is a large number of such constraints and challenges for inter-UAV communication; we refer to [43] for an overview of the literature.

### 2.1.5. Summary

The Search and Rescue (SAR) scenario is depicted in Figure 2, where the agents maneuver a complex indoor setting to reach a target, partially guiding the search using a beacon signal. Once the location of the target has been determined, the scattered agents realign to form the shortest possible relay link back to the starting point (base station). All unused agents return to the base to prevent over-utilization of resources.

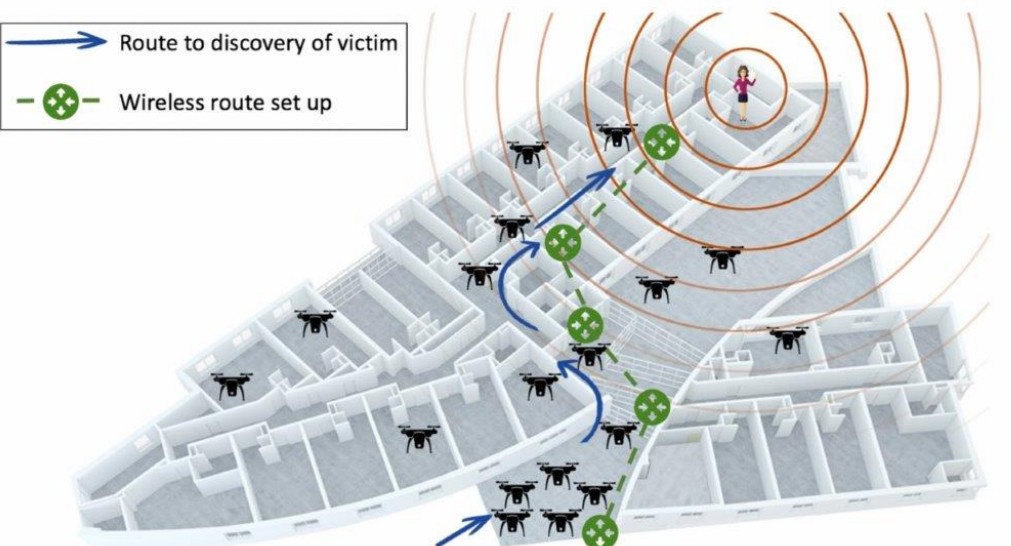

**Figure 2.** An indoor Search and Rescue (SAR) scenario where the victim sends out a distress signal to guide a search conducted by autonomous robotic agents. On discovery of the victim, the agents create an optimal relay link (by optimizing various aspects such as time, cost, length of the link, etc.) to connect the victim to the base station. The other scattered agents not involved in the relay can be called back to the base.

### 2.2. Wireless Sensor Networks (WSN) in SAR for Swarms of Drones

#### 2.2.1. Mobile Wireless Networks

Mobile wireless networks, i.e., networks where at least some of the nodes are mobile, have the potential to temporarily upgrade existing IT and communication infrastructure as and where needed [44–47].

Communication networks have to be dimensioned to accommodate peak demands, which implies that outside those periods, there is redundancy in the system. Given this redundancy, the ability to dynamically allocate additional capacity can dramatically increase the cost efficiency of the utilized resources. Due to this, telecom operators around the globe regularly deploy mobile cell towers during large public events (e.g., sporting events) or during times of known or predicted peak demand (e.g., New Year's Eve).

#### 2.2.2. Mobile Wireless Sensor Networks (MWSN) in SAR

The use of mobile infrastructure is not restricted to delivering communication services, as e.g., surveillance infrastructure (e.g., CCTV cameras or crowd sensors) is commonly deployed alongside the above-mentioned communication infrastructure. In case of an emergency situation, especially in SAR operations, fast, cost-efficient and simple deployment is a crucial factor when it comes to saving lives and reducing economic damage [48]. When the environment under considerations may have been altered or have suffered damage, the ability to deploy sensing capabilities alongside other equipment may facilitate real-time planning based on the gathered information. Generally speaking, the more adaptive and reactive a deployment process into such an environment is, the more efficient and safe it can be. In the literature, the principle of self-organization is increasingly applied to such networks due to the fact that decentralized operations further reduce the need for communication infrastructure and speed up the decision making.

#### 2.2.3. UAVs as Nodes in Mobile Networks

The use of UAVs seems a straightforward consideration, and indeed, there is a host of literature and surveys [49] on the use of UAVs as mobile nodes in communication networks [50], or as parts of dynamic networks [51]. Practitioners in the field have identified UAVs, used as mobile sensing devices, as having great potential when it comes to

providing communication support [35,52] and mobile communication infrastructure [9,18], such as a drone-based (mobile) wireless sensor network (WSN) [53] or a wireless local area network (WLAN) in which communication is realized node to node (N2N) using ad hoc [54] routing [55] to exchange messages between nodes. Examples for the use of drone-based wireless sensors networks (WSN) [56–58] are, e.g., healthcare and environmental monitoring [11,59–61]) as well as reconnaissance in the aftermath of a disaster [10], and general applications where the topology of the network is dynamic and adaptive [62].

### 2.3. Nature-Inspired Approaches to Finding the Shortest Path

Cooperation and self-organizing [63] collective behavior has been observed in group-living animals from insects to vertebrates, as well as at cell level [64]. Engineers and computer scientists have studied the underlying guiding principles and successfully used them to increase the accuracy of models and to improve algorithms (cf. [65]). This has the potential to be extremely efficient [66]. One of the defining characteristics of approaches based on, e.g., the foraging behavior of social insects is their distributed nature. Distributed collective sensing has been shown to be possible using only rudimentary cognition, and is cost effective for computational agents [67]. Global goals can be achieved by locally processing the agents' (limited) view of the world [68]. To this end, computer scientists have studied models from theoretical biology and applied the underlying principles to a variety of computationally hard [69] problems, often in the form of heuristics.

Computer scientists have studied models from theoretical biology and applied the underlying principles to a variety of problems, often in the form of so-called *heuristics*; that is, approaches that *find* (as opposed to *deriving deterministically and exhaustively*) good (as opposed to *the best possible*) solutions to problems. Heuristics [70] have been used to find *very good* solutions in an extremely short time (e.g., approaches where the computational cost increases only linearly with the size of the problem). For a broad overview and a useful compilation of nature-inspired approaches (including the algorithms), we suggest [71], which is also available online for free (http://www.cleveralgorithms.com/, accessed on 1 August 2022).

#### 2.3.1. Deterministic versus Heuristic Search

Deterministic techniques have been the conventional approach to path planning in some approaches [42,72]. Some prominent techniques include searches on Visibility Graphs (VG) and Voronoi Diagrams (VD) [73], Cell Decomposition Method [42], and gradient techniques such as the Artificial Potential Field Method [74–76], amongst others. Deterministic techniques, such as the simple wall-follower algorithm, work optimally in maze search problems, provided the algorithm has complete knowledge of the search space beforehand [77]. When used in dynamic or unknown environments, deterministic techniques could end up in infinite loops or stuck in local optima, which could impede performance in SAR operations. Table 1 summarizes the research in the field of maze solving using deterministic and heuristic approaches for both local and global path planning.

**Table 1.** Research in the field of deterministic and heuristic path planning strategies in maze-like environments, including Ant Colony Optimization (ACO), Particle Swarm optimization (PSO), and others. **LP** stands for *Local Planning*, **HT** stands for *Heuristic Technique*.

| Ref | LP | HT | Algorithm Used |
|:---:|:---:|:---:|:---:|
| [78] | | ✓ | Improved ACO |
| [79] | ✓ | | Artificial Potential Field Method |
| [80] | ✓ | ✓ | Multi Pheromones for tracking targets |
| [81] | ✓ | ✓ | Improved ACO Heuristic Function |
| [82] | | ✓ | Simple ACO |
| [83] | ✓ | ✓ | Repellent Pheromone for coverage |
| [84] | ✓ | ✓ | Advanced PSO |
| [85] | | ✓ | Genetic Algorithm with ACO |
| [86] | | ✓ | Hybrid ACO w. Random- + RL based-Search |
| [87] | ✓ | | Point Bug Algorithm |
| [88] | ✓ | | Dijkstra Algorithm |
| [89] | ✓ | ✓ | Fuzzy Logic with Counter ACO |
| [90] | | ✓ | PSO in partially known environments |
| [91] | ✓ | ✓ | Combination of multiple pheromones |

2.3.2. The Dijkstra Algorithm

Dijkstra's algorithm is a well-known search algorithm developed by a Dutch computer scientist, Edsger Dijkstra, in 1959 [92], which can be applied on a graph. The algorithm is one of the solutions on Single-Source Shortest Path problem (SSSP) with directed or undirected graphs, and has edges with non-negative weights [93]. The SSSP is defined as finding the shortest path from a distinguished vertex in a graph called source to every other vertex in the graph. Dijkstra's algorithm follows the greedy approach, which means it finds an optimal path with the least total cost (shortest path from source to destination) [94].

The algorithm begins with a graph with nodes, $u$ or $v$, weighted edges connecting nodes denoted as $(u, v)$ and weights denoted as *CostMatrix(u,v)*. The initiation of values and related steps before starting the path finding are:

- An array holding all edges costs (*distance(i)*) where all values are initiated to infinity except the first value (*distance(source)*), which is set to zero.
- An array that contains all the nodes that have been visited during the search, which, by the end, contains all nodes in the graph (denoted as *visited*). Then, the algorithm proceeds as follows:
- While the visited array does not contain all nodes, we take node $v$ with the least *distance(v)*. Initially, it will be the source because *distance(source)* is set to zero.
- Node $v$ is then added to the visited array, indicating that it has been visited.
- Update distance values of adjacent nodes ($u$) to the node $v$.
- If

$$distance(v) + CostMatrix(u,v) < distance(u),$$

then there is a new minimal distance founded for $u$, so *distance(u)* is updated with the new minimal value. Otherwise, no changes are made to *distance(u)*. Finally, after the algorithm has visited all nodes in the graph and the smallest distance to each node is found, distance will now contain the shortest path tree from the source node [95].

Dijkstra's main advantage is that it finds the most optimal path in an efficient manner. However, it may incur a large cost in computational time due to the blind search it performs to calculate the optimal path. Another weakness is that the algorithm does not consider negative edges. However, in this project, the distance is seen as the cost with positive value, which means that only positive edges are considered, following [92]. A brief summary of examples in the literature on using Dijkstra is shown in Table 2.

**Table 2.** Research using deterministic and heuristic path-planning strategies; (✓) indicates maze-like environments with obstacles. The used algorithms are **D**ijsktra (**D**), **F**loyd-**D**ijsktra (**FD**), **A**nt **C**olony **O**ptimization (**ACO**); *others* refers to a combination of sweep and savings.

|  | Reference | Cost Function | Alg. |
|---|---|---|---|
| ✓ | [93] | Distance, Gas Concentration | **D, ACO** |
|  | [96] | Energy | **D** |
|  | [97] | Distance | **D** |
| ✓ | [98] | Energy, Difficulty, Distance | **D** |
| ✓ | [99] | Distance | **FD** |
| ✓ | [100] | Distance | **D, ACO** |
| ✓ | [101] | Distance | **D + others** |

2.3.3. Ant Colony Optimization (ACO)

In nature, several insects and animals have an inert sense of swarming together to accomplish a task more efficiently. Swarming, when used as a means for collaboration, can help optimize both the time and the energy spent. Adding this feature to our research approach would be beneficial, as it has the same setup of a search in an unknown environment guided by an external stimulus. The stimulus, which in the natural swarming process may be e.g., the smell of food, is in our case the beacon (cf. Section 3.3.1 for details).

Ant systems are an example of accomplishing a task by collaborating via *stigmergy*, i.e., using the environment to guide the actions of the individual. Ants effectively use the world as a sort of shared memory. They do so by depositing a trail of pheromones on their way. When using the aggregated pheromone levels in their environment to influence their path planning, the collective manages to converge on the shortest path to the food source, despite all obstacles, and can possibly do so in optimal search time.

Ant travel is generally a positive feedback mechanism. An ant's decision at every fork is affected by its likelihood to adhere to paths with more pheromone concentration, in addition to a random element. This random element encourages exploration of other less-traveled trails that could possibly lead to better solutions. Furthermore, the pheromone also evaporates over time. So, in the context of a newly emerged shorter path to the food source, most ants would slowly migrate to the new path due to the decay of pheromone and the new pheromone deposits. For a simulation of pheromone-led ant travel, the probability of an ant taking a particular path is represented by its density function, which is given by

$$P_A(t+1) = \frac{(c + n_A(t))^\alpha}{(c + n_A(t))^\alpha + ((c + n_B(t))^\alpha} \tag{1}$$

where $c$ is the degree of attraction to an unexplored path (the random element), $\alpha$ is the bias to using a pheromone concentrated path, and $A$ and $B$ are the two paths to choose from at a fork [102]. Given the specific nature of the application we discuss in this article, we have explored the ranges of values for $\alpha$ and $c$, namely alpha in the range $[1.8, 2.2]$, and $c$ in the range $[18, 21]$ A parameter space exploration (omitted here) demonstrated empirically that $\alpha \approx 2$ and $c \approx 20$ provide the best fit to experimentally observed behavior.

**3. Search and Rescue in Maze-like Environments**

*3.1. Modeling the Problem*

3.1.1. Indoor Search and Rescue Operations: A Maze Exploration Problem

By the very nature of the intended application domain, the environment for an indoor search-and-rescue operation is (at least partly) unknown. The problem of searching for a victim (locating a specific position in the environment, reaching it and maintaining and updating the path from that position to return to the point of entry) in such an environment is often, and aptly, modeled as a maze-solving problem. The randomness of the maze walls can represent the random placement of obstacles in the area of interest. Therefore, the task

of maze solving is essentially a path-planning problem in an unknown environment. The main issues associated with maze solving include redundancy in the found path [40,41,86], and premature convergence while finding local optima [74,103]. Several studies have attacked these problems using different path optimization techniques.

Therefore, we used a collection of mazes with different layouts, sizes and complexities. The mathematical model and a measure capturing the complexity of these mazes are presented in Section 4.1.1.

### 3.1.2. The Physical Accessibility

Mazes as a concept are rich with historical connotations, but abstractly speaking can be seen as environments with obstacles, commonly oriented such that elongated hallways or paths are created. In contrast to a labyrinth, mazes have branching points; that is, there are locations where the next step (which does not reverse the previous step, i.e., going back) can be in more than one direction. This gives rise to the most commonly known property of mazes, namely that the finding of a specific location, or retracing one's steps to return to the entry point, can be challenging. While there are many variations on this theme, in the simplest form (and, as it turns out, a sufficiently general description) a maze is a collection of locations combined with a accessibility relation that determines, for any two locations, whether they are directly connected. Due to this, mazes can be represented as graphs [104].

### 3.1.3. The Signal Accessibility

Contrary to the impact walls have on the physical accessibility to locations for, e.g., humans (which is boolean: either possible or not possible), walls have a more complicated effect on e.g., radio waves. Radio signals, while also affected by the physical barriers, can penetrate walls and other objects to a certain extent. In the absence of blocking obstacles, a radio signal can be expected to extend outwards from the source homogeneously, with its strength decreasing with distance to the signal source or beacon. Figure 3 visualizes this for a specific scenario and beacon as well as for specific properties of the walls in our scenario. The mathematical model for the signal strength and propagation is provided in Section 4.1.2 by Equation (5).

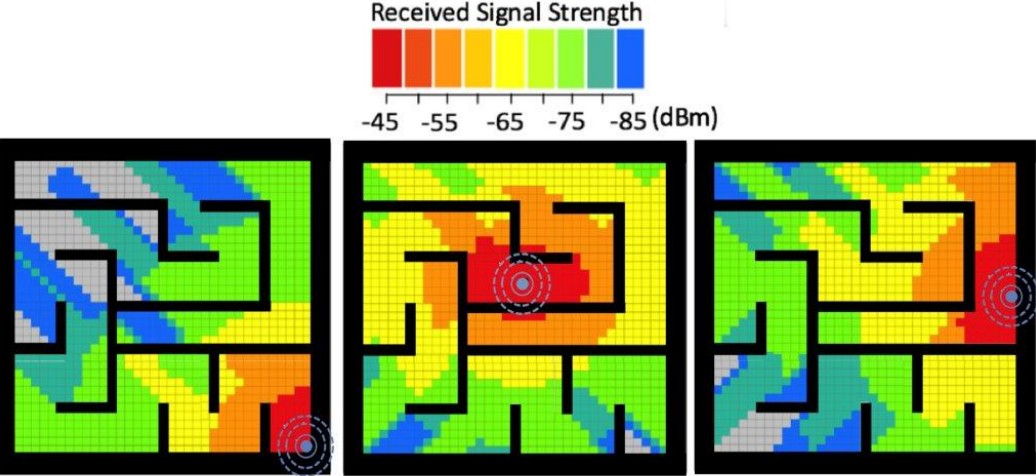

**Figure 3.** Color-coded illustration of the beacon signal power received across the maze. The signal attenuates with distance and across the obstacles (walls), shown here in black color. The attenuation is a function of the distance. The signal undergoes added attenuation across walls due to shadowing.

### 3.1.4. Modeling the Drones

At the present stage of the development and application of this hybrid model, we only account for a controlled discretized motion of UAVs, as specified, for example, in Table 4. While this obviously ignores virtually all aspects of actually deploying drones inside SAR

environments, it suffices here to provide a minimally adequate model for the operation of UAVs and allows us to use our algorithms to drive the movement of the drones, albeit restricted to the choices *hover (do not move)*, *turn clockwise*, *move forward* or *move backward*.

### 3.2. Search, and Rescue: A Problem of Two Sequential Phases

Figure 4 depicts the maze-solving equivalent of the situation in Figure 2, where the target is assumed to be at the "end", or a far corner of the maze at the exact opposite of the start point for the agents; Figure 4(1). The agents are required to solve the maze (Figure 4(2)) and set up a reverse route (Figure 4(3)), without any prior knowledge of its topology. This is the analogy that will be used throughout the course of this paper. In the absence of a common benchmark case, several tests are carried out by varying different aspects of the maze, such as maze dimensions and complexity, and the size of the search group.

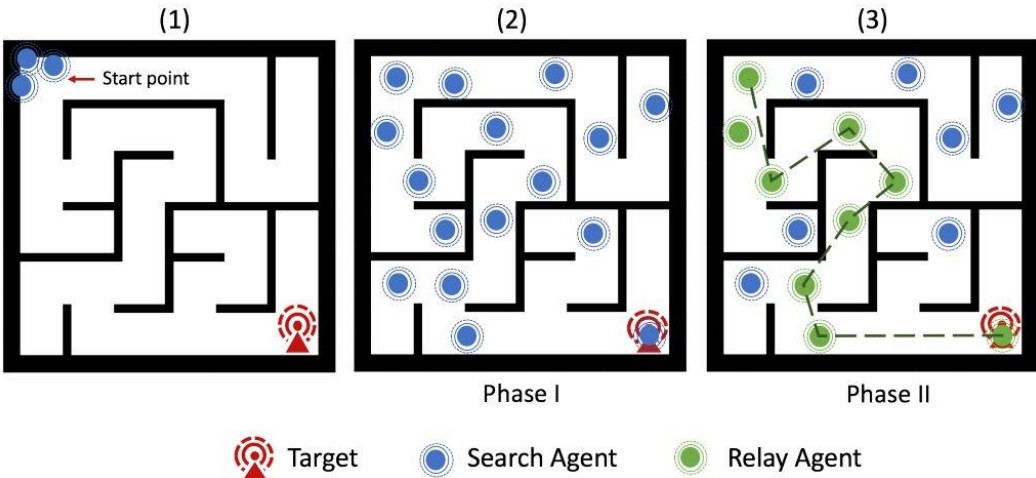

**Figure 4.** The indoor search is translated into a maze-solving problem. Just as with the indoor search introduced in Figure 2, the agents enter the maze at a specific start location / point (1), traverse the maze in search of the target (depicting the victim) in Phase I, shown in (2), and then (in Phase II) calculate the shortest and cheapest route amongst the scattered agents (3) to form a relay network.

Therefore, the SAR operation is translated into a two-phase process, where the search agents first collaboratively search the area to locate the victim, and upon discovery, shift to relay mode, where a temporary relay network is set up using a subset of the originally scattered search agents, to complete the rescue operation. Figure 5 shows the flow of control between the different sections of the algorithm through the search process.

### 3.2.1. Phase I (Search): Maze Exploration

For the purpose of designing a path planning strategy for SAR, our focus is not on finding the absolute shortest path, but rather a good (but possibly sub-optimal) path, and to do so in the shortest time possible. This is to reduce the response time. The path could be further shortened in Phase II (Rescue), while reverse routing in the search phase.

Our ultimate focus is on the search time, rather than path length. For this purpose, heuristic techniques are favorable due to their element of randomness, which pushes the agents to explore while still partially following an order.

Therefore, in Phase I, a swarm of robots are injected into the search space. Each robot marks its explored territory in pheromone maps, local copies of which are shared between robots within a communication range. These pheromone map updates signal other robots to avoid already mapped/explored areas, quickening the overall exploration while shortening the search time.

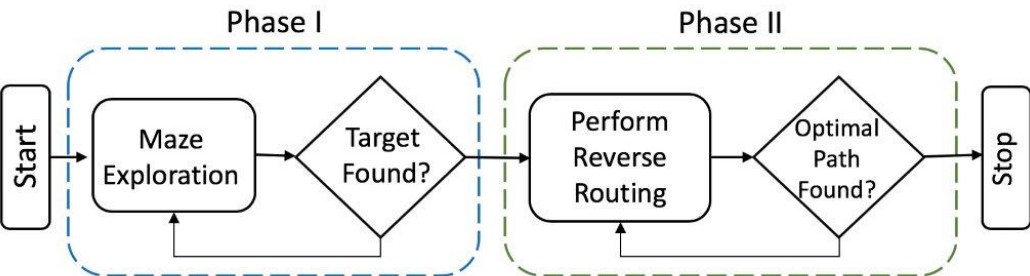

**Figure 5.** The search and rescue process can be seen as two sequential phases: Phase I, where the search is conducted in an environment; and Phase II, where the extraction of an item or person from the environment is realized (i.e., the rescue part). For details, see Section 3.2.

### 3.2.2. Phase II (Rescue): Signal Routing and Victim Extraction

Once the agents find the victim, they should stop the exploration and the agent at the target should find a relaying route leading back to the source node in order to pass the target's information. This is done by finding and setting up the shortest possible link between the agents that are injected in the area. Among the possible methods used to find the most optimal path, we employ a simple variant of the Dijkstra Algorithm that uses the distance as the cost. As a route is calculated backwards to the source, the selected next-hop agents are kept fixed in position to form a relay network to assist the rescue operation.

### 3.3. Solving Phase I (Search): Maze Exploration

The first phase of the Search and Rescue operation starts with the search agents rapidly exploring the maze, partially guided by the strength of the received beacon signal, as well as being influenced by the movement of fellow agents.

#### 3.3.1. The Approach

A general procedure for the implementation of an ant system was introduced in Section 2.3.3. The AA local path-planning-based algorithm applies the ACO probability density function at each step to determine the next best step using:

$$P_i(t+1) = \frac{(c + n_i(t))^\alpha}{\sum_{j=1}^{5}(c + n_j(t))^\alpha} \tag{2}$$

where $P_i(t+1)$ is the probability of moving in direction $i$, and $n_i(t)$ is the amount of pheromone in block $i$. The summation ($j$) in Equation (2) is over the five possible directions of movement (cf. Figure 6) with the 5th direction remaining stationary. Additionally, $c$ is the degree of attraction to an unexplored path, and $\alpha$ is the bias in using a pheromone-concentrated path. An example of AA parameter values would be $c = 20$, and $\alpha = 2$, as demonstrated in [105].

With AA, however, ants were noticed to be clustering together, due to the attractive nature of pheromone which makes ants follow each other with little randomness. To counter this effect, and to quicken exploration, an inverted AA (iAA) model was also developed, where the pheromone was designed to be repulsive, encouraging ants to venture into unexplored territory, by making the attraction component negative. The probability density function used for decision making in iAA is

$$P_i(t+1) = \frac{(c + n_i(t))^{-\alpha}}{\sum_{j=1}^{5}(c + n_j(t))^{-\alpha}} \tag{3}$$

With promising initial results, two new versions of iAA, namely inverted-AA with beacon initialization (iAA-B) and inverted-AA with an increased sensing range (iAA-R), were also developed. The iAA-R was developed to test the effect of a longer sensing range on the

speed of convergence of the solution. The workings of a basic AA model are shown in Algorithm 1. The different models differ only in terms of the Probability Density Function used to generate the roulette wheel distribution in the algorithm. A visual depiction of the decision making based on the pheromone levels for all four models is shown in Figure 6.

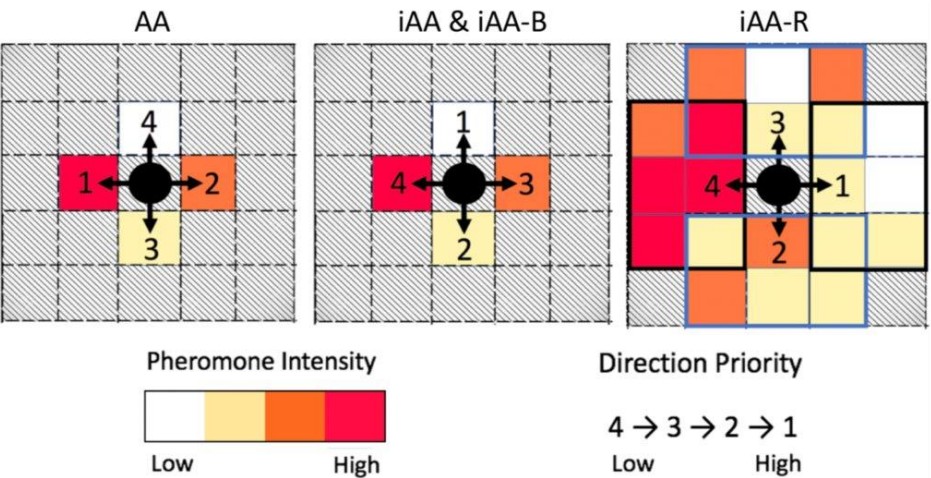

**Figure 6.** All AA-based decision-making models prioritize their 5 possible directions of movement (1–4 are shown above, with the option to remain stationary being the 5th). AA prioritizes moving to a cell with a higher pheromone level, while all iAA-based models prefer moving to a cell with a lower pheromone concentration. The iAA-R algorithm (rightmost image, above) checks for average pheromone in neighboring *regions*, while the other three approaches (AA, iAA, and iAA-B) only check pheromone levels in the immediate neighboring *cells*.

### 3.3.2. The Algorithm

The pheromone used to implement the AA-based algorithms is virtual, stored as matrices in individual agents. The virtual pheromone, therefore, is a message-passing coordination mechanism between the agents in the system. The pheromone matrix helps the agents keep track of the pheromone intensities, while at the same time, it helps them roughly map out the explored area.

---

**Algorithm 1:** The basic AA Algorithm used for decision making, where each model differs in probability generation. The AA, iAA, iAA-B, and iAA-R models are based on the same algorithm; they differ mainly in the equation used for probability distribution generation.

---

initialization;
possible moves = [stay, right, left, forward, backward];
**while** *target not found* **do**
  **for** *each ant* **do**
    │ Generate list of all possible next states;
    │ Acquire pheromone information of all next states;
    │ Roulette Wheel ← generates probabilities of moving to each next state;
    │ Spin Roulette Wheel to pick next state;
    │ Update current position and pheromone levels;
  **end**
**end**

---

As each agent explores the search space individually, their pheromone matrices would differ. Hence, when two agents pass by each other, they exchange their copies of the matrix to combine their search results and come up with a more inclusive version of the search space and pheromone intensities. For a general *"bird's eye"* view of the process, the depiction of the search is shown in Figure 7.

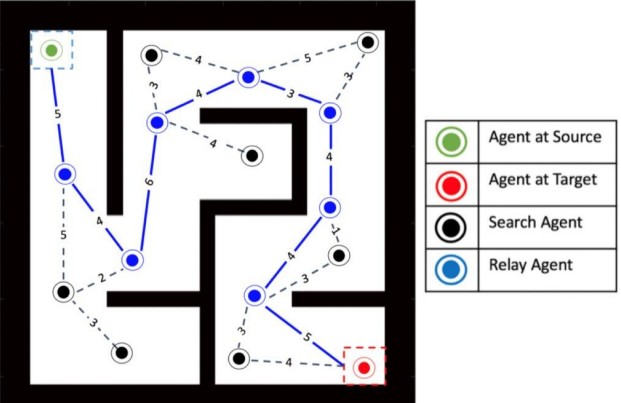

**Figure 7.** The path cost calculations, having found an optimal path back to the source. At each hop, the algorithm favors the link with the lowest cost associated with it (indicated by the integers).

### 3.4. Solving Phase II (Rescue): Signal Routing and Victim Extraction

In a Search and Rescue, once the trapped person (or the target) is located, the MANet nodes reorganize themselves into a relay network to inform the base station of the target coordinates and call for help. In this phase, for computation purposes, the scattered MANet nodes can be treated like a graph $G$: a data structure consisting of two sets $(V, E)$, where $V$ is the set of vertices and $E$ is the set of edges connecting any two vertices in the graph.

A graph is usually represented as a diagram where vertices are symbolized as points and edges as lines connecting its end vertices [106].

Graphs have two main traversal approaches: Depth First Search (DFS) and Breadth First Search (BFS). DFS approach starts graph traversing at an initial node, which is called a root, and then goes down one path (branch) till it reaches the end of it. Then, it backtracks until it reaches the root and chooses another path to explore it. This is repeated until the graph is fully traversed. On the other hand, BFS starts at an initial root node then moves layer by layer in the graph, where a node is explored first with its neighbors, then the node is explored in the next layer. These layers are called depth levels [106].

#### 3.4.1. The Approach

A graph is formed after exploration with agents as nodes, and the edges are links between agents. The links between agents are present if two agents are within each other's Line-of-Sight and Transmission Range. Using the graph formed, an adjacency matrix is then produced that has the weights of the edges in the produced graph, which is called the *Cost Matrix*. The weights of the edges are evaluated based on the Euclidean distance between two agents ($u$ and $v$), defined in the following equation:

$$EuclideanDistance(u, v) = \sqrt{(x_u - x_v)^2 + (y_u - y_v)^2} \qquad (4)$$

In an ideal scenario, with a successful conclusion of Phase I, the agents would be already interspersed in the maze (or search space), with no big voids or obstacles between them. In this circumstance, a path back to the origin can be easily found using the classic Dijkstra algorithm. However, the presence of obstacles or non-uniformity in the arrangement of the agents, in the form of large voids that disrupt communication links, a position-based algorithm is proposed to find a feasible complete path. The algorithm focuses on stopping the agents in the right locations to achieve a better spread of the agents in the environment.

#### 3.4.2. The Algorithm

In the proposed approach (cf. Algorithm 2), we make use of a *"depth attribute"* to keep track of the diffusion of agents through the maze in order to find a path to the source. To do so, an individual measure of depth is calculated. In a tree data structure, the depth of a node is simply the number of edges from the tree's root node to the node itself [107] and

therefore easily calculated. The root node is the lowest level of the tree and is assigned depth 0. The next level, connected with only one edge to the root, has depth 1. As the connections to the root grow, so does the level, with the final level (here) being 3.

---

**Algorithm 2:** Algorithm to find Shortest Path to Source from Target

---

    *Last_Stopped* = First agents to arrive at target;
    *Stopped_Agents* = [*Last_Stopped*];
    *Possible_Moves* = [right, left, forward, backward];
    **while** *Source not within Stooped_Agents* **do**
        **for** *each agent not in Stopped_Agents* **do**
            Generate list of all possible next states;
            Acquire pheromone information of all next states;
            Roulette Wheel ← generate probabilities of moving to each next state;
            Spin Roulette Wheel to pick next state;
            Update current position and pheromone levels;
            *Agents_Nearby* = [All agents that could connect to *Last_Stopped*];
            *Last_Stopped* = Min(depths of *Agents_Nearby*);
            *Stopped_Agents.push*(*Last_Stopped*);
        **end**
    **end**
    end function;

---

In the maze-routing problem, the same concept can be applied, where the tree's root is the source and the depth there will have the value 0. As the agents move away from the point of origin, some of them will lose direct access to the source, either because of maximum range constraint or because the line of sight may be broken. However, agents will still remain in contact with other agents in the maze. Then, a rule will be followed: out of all the agents around that the agent can connect to, it chooses the one with the lowest depth, and then sets its own depth to this $(depth + 1)$. If all agents concurrently and iteratively apply this rule, then their individual depth should remain a measure of the number of hops to the entry point (and the path will be explicitly identified).

Figure 8 shows a flowchart of the working of the system in Phase II. The *Last_stopped* agent shown, is the one that was last chosen to be stopped by a previous stopped node and has to choose an agent from the nearby set of agents, subject to:

- It is within the transmission range (*TRange*) of the *Last_stopped* Agent.
- There are no obstacles in between it and the *Last_stopped*.

In the scenario shown in Figure 8, from the set of *Nearby_Nodes*, the first agent has depth of 0 which means that it can see the source. The other agent has depth of 1, which means that its one hop away from the source. Therefore, from the set, the next stopped agent to be chosen is the one with the lowest depth, which is the agent with depth 0. Moreover, after the node is stopped and because it is within the source transmission range and it does not have obstacles blocking it, the proposed search algorithm ends and Dijkstra algorithm runs again to find an optimal shortest path to the source. This will help in making sure that a path to the source can be efficiently formed after running Dijkstra algorithm.

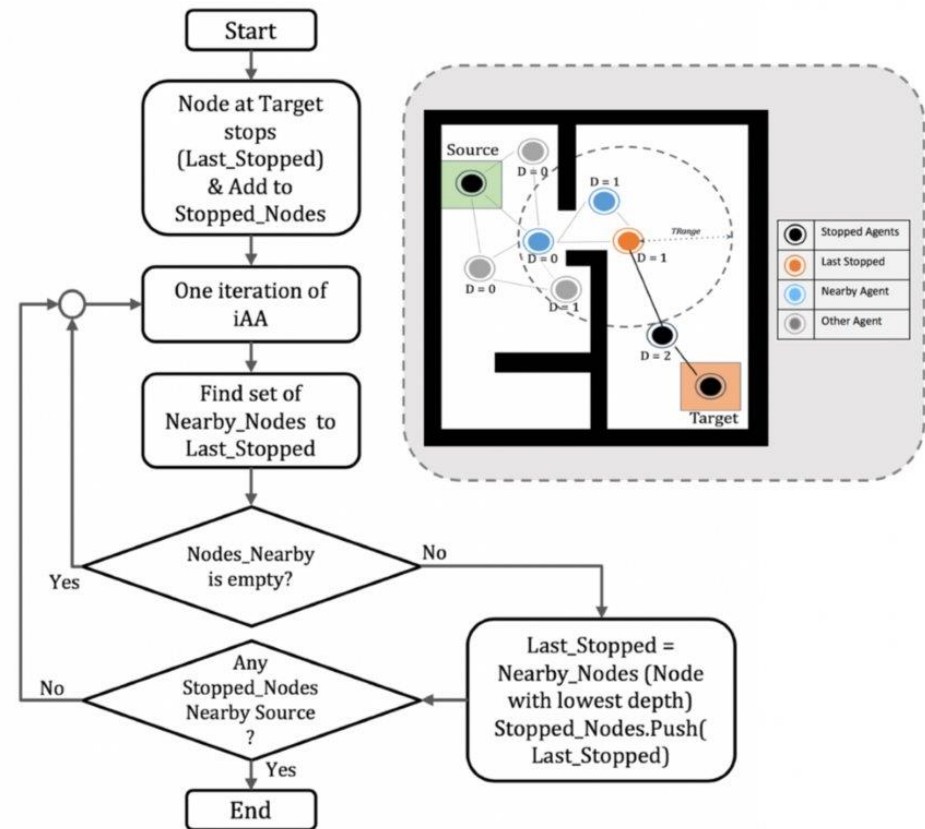

**Figure 8.** Flowchart describing the working of the system in phase II. The last stopped node considers only the nodes within its communication range as its neighbors in the calculations. If there are no nodes within its communication range, the iAA simulation is restarted to move the nodes until at least one neighbor appears.

The solution proposed is considered a semi-distributed model. At first, the agents move in a distributed manner for the exploration part, then when the target is located, the agents will broadcast a message that the target is located back to the source, which will then find the shortest path in a centralized fashion. If no path is constructed, at first the agents will continue with the proposed improvement that is distributed. Agents will communicate when the target is found, and then the graph is constructed where the source and the target are included in it. If not, then the agent at the target tries to construct a path by searching for agents nearby until the source is part of this path. Finally, the source can communicate to agents which agents form the optimal path. Figure 8 explains how the agents communicate when the target is located.

### 3.4.3. Routing

For a simple simulation, we can assume the walls are impenetrable to the signal exchanged between the agents, and that the agent should have a line-of-sight requirement to establish a connection to other agents. Figure 9 shows a scenario where no path is constructed after reaching the target because there is no subgraph connecting the source to the target. Figure 9 (1) shows the state where the target is found (node circled in green); however, when calculating the distances, which is the cost matrix, and finding possible connections, one can see that there is no path found between source and target nodes.

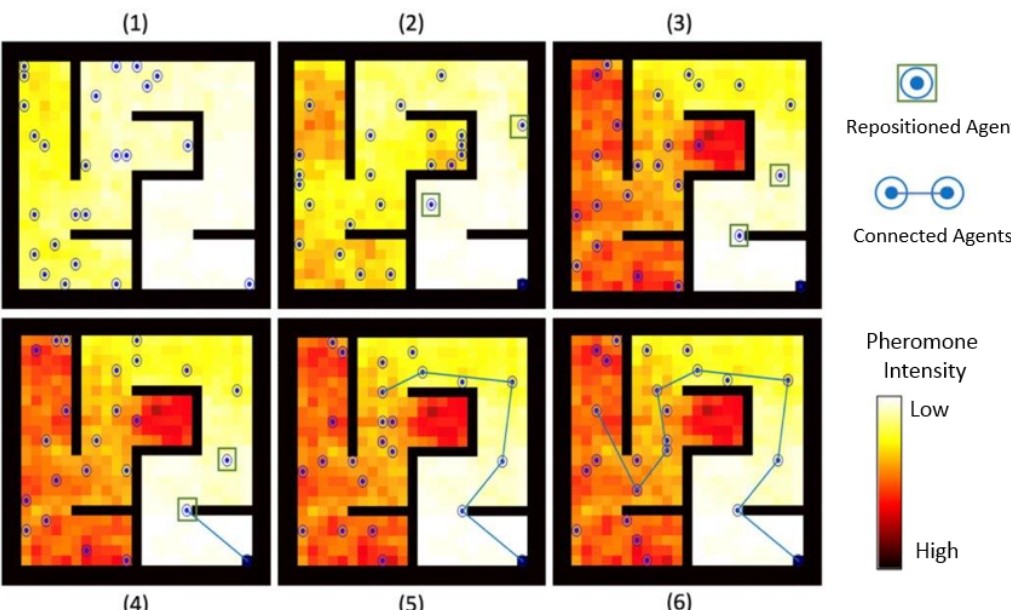

**Figure 9.** A visual example showing the proposed approach: in (1) the target is located and the agent stops and starts looking for nearby agents, which (2) continue exploring. In (3) one agent is nearby, in (4) a link to that agent is created, and it stops. This process continues (5) to other agents, using the least depth as the guiding factor until (6) the process ends when the reached agent is near the source, causing the Dijkstra algorithm to terminate.

The exploration continues using the iAA, except for the node at target, as it stops and tries to connect to nearby nodes, as one can see in Figure 9 (2). When a node is near to the stopped node, as seen in Figure 9 (3), it is able to connect to it, and it stops it. Now, the last stopped node starts looking for the nearby nodes. Then, the model continues doing the same with every node connected in the subgraph until one of the nodes is near to the source, as seen in Figure 9 (6). After reaching this state, Dijkstra re-runs, as one is sure that a path will be created after stopping nodes in the right locations.

Figure 10 shows the situation of the agents before (left panels in both Figure 10a and Figure 10b) and after (corresponding right panels) routing has been performed. In the right panels, the agents marked in red have become part of the developed relay network. The remaining agents can be called back to the base station without affecting the performance of the relay.

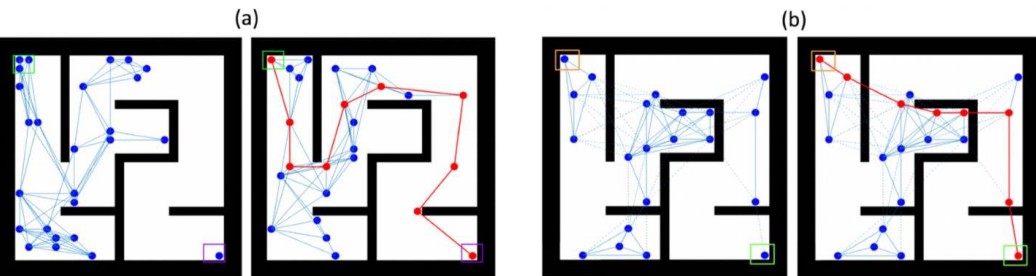

**Figure 10.** Reverse route set up using the modified Dijkstra approach with implementation of both (**a**) impenetrable and (**b**) penetrable walls. The path length in the case of penetrable walls is much shorter than the former. However, this is heavily dependent on the material of the walls in the search space, which shows it could have a heavy influence on the performance of our technique.

## 4. Materials and Methods

### 4.1. Modeling Choices

4.1.1. Maze Size and Complexity

As stated above, we model a collection of mazes with different layouts, sizes and complexities. To simplify the computation and implementation of the system, the mazes were discretized using a grid method similar to the one in [72]. The maze is decomposed into a grid, where the unit grid size is the same as the agent size, thereby making the step size equal to a unit grid. This discretization helps visualize the system as a matrix, which makes computation of signal reception, agent localization and move generation easier. Naturally, future work would address the situations in which the size of the agent and the size of the grid unit are not equal. Figure 11 shows one sample of each of the five complexities (listed in Table 3) of the mazes used in the tests.

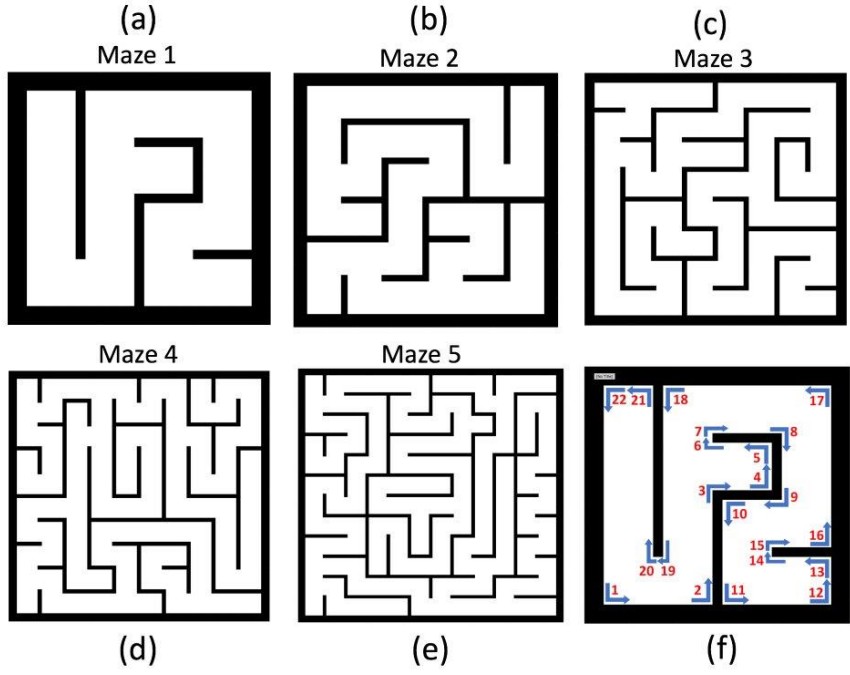

**Figure 11.** (**a**–**e**) show the complexities of the 5 sample mazes (henceforth: M1-M5) referenced in this paper, cf. Table 3 for a numerical comparison. These are provided as examples. In total, 10 different layouts were generated for each complexity listed in Table 3 to further gauge the affect of randomness on the results. Maze (**b**) was previously used in Figure 4. The last panel, (**f**), illustrates how we calculate maze complexity, using the example of maze (**a**): the complexity is the number of 90 degree turns an agent can make in the maze, cf. (**f**).

For each of the five types of mazes, 10 different layouts were randomly generated. Table 3 summarizes these maze sizes and provides a comparison of the respective complexities. Here, we rely on a simple technique of counting the number of 90 degree turns an agents can make at the corners in the maze as a measure of the maze's complexity.

**Table 3.** Dimensions and average complexities of the 5 maze types used in simulations; see Figure 11.

|  | Dimensions | Complexity |
|---|---|---|
| Maze 1 (M1) | $27 \times 27$ | 22 |
| Maze 2 (M2) | $39 \times 39$ | 47 |
| Maze 3 (M3) | $51 \times 51$ | 81 |
| Maze 4 (M4) | $63 \times 63$ | 129 |
| Maze 5 (M5) | $75 \times 75$ | 210 |

### 4.1.2. Signal Progression through Obstacles

To model indoor signal propagation, the ITU model, proposed by the International Telecommunication Union, described in [108], was modified through the addition of the $(w \times c)$ term to account for attenuation caused by the internal walls/obstacles to the following form:

$$PL[dB] = 20log_{10}(f) + 28log_{10}(d) - 28 + (w \times c) \tag{5}$$

where $w = 4.4349$ is the wall attenuation factor (for brick wall) [109], $c$ is the wall count encountered, $f = 2400$ MHz is the frequency channel of communication for standard Wi-Fi, and $d$ is the distance, in meters, from the beacon source [108]. Figure 3 shows the signal propagation with a corner-located beacon.

In real-life situations, a radio wave is usually able to propagate through obstacles (depending on the thickness and material of the obstacle), but the signal strength significantly reduces due to a phenomenon called *path loss*, for which we use the ITU model to model an indoor beacon propagation. This path loss can, in effect, be simulated by artificially lengthening a link across a wall. Accordingly, Equation (5) is rewritten to calculate the re-interpreted distance between the agents as:

$$Distance(m) = 10^{\left(\frac{PL+28-20log_{10}(f)}{20}\right)} \tag{6}$$

This value is then fed into the *Cost Matrix* (introduced in a simplified form in Section 3.4.1). Due to this, the *Cost Matrix* represents the new weight of the edge between two nodes and the Dijkstra algorithm, as we already described, is run to find the shortest path from the target back to the source, with the possibility that the signal will penetrate the walls. A visual representation of how walls can affect signal strength is shown in Figure 3; the impact of allowing sufficiently strong signals into the path even though the signal goes through a wall can be seen in Figure 10, where the top version does not include signals that pass through walls, while the bottom does.

### 4.2. Data Collection
### 4.2.1. Hardware/Software Used

All computation is carried out using MATLAB 2018b; results are later checked with subsequent versions of MATLAB. The codes were developed on a laptop running Windows 10, with 16 GB RAM. The simulation code executes on a 1–12 min scale, depending on the number of agents used, and on environmental parameters for Search and Rescue.

### 4.2.2. Maze Complexity

Searching in an obstacle course such as a maze with a heuristic technique such as an Ant Algorithm is a process *"rich in randomness"*, leading to excessively random search statistics (in terms of run time, number of steps, etc.). Therefore, in order to properly assess the performance of our implemented techniques, we carry out simulations on a collection of mazes with different layouts, sizes and complexities, each with 50 repetitions averaged out to realize the extent and appropriateness of the variations in the results. Figure 11 shows the five different maze complexities that were simulated, with Table 3 (on the same page) providing a comparison of the complexity values for these types. For each of these five types, a total of 10 different layouts were generated to further gauge the affect of the randomness.

### 4.3. Comparative Evaluation of the Algorithms

To study the impact of AA on the autonomous ants-led search in the maze, a simple ACO-based algorithm and a standard random search were implemented for comparison. A set of experiments were set up to simulate AA and iAA algorithms, with variable group sizes ranging between 100 and 600, with 100-unit increments.

Both plots in Figure 12 visualize the trend of the energy expenditure for 5 different maze complexities (measured as shown in Figure 11, in terms of 90 degree turns encoun-

tered) of 22, 47, 81, 129 and 210 (cf. Table 3). There is a 30% average increase in energy expenditure in solving the least and most complex mazes for the same group size. The decrease in the number of steps needed to solve the maze is more obvious in the larger mazes, with an approximate 27% decrease in the largest maze. The energy expenditure increases almost linearly with the increase in the number of agents in the search group.

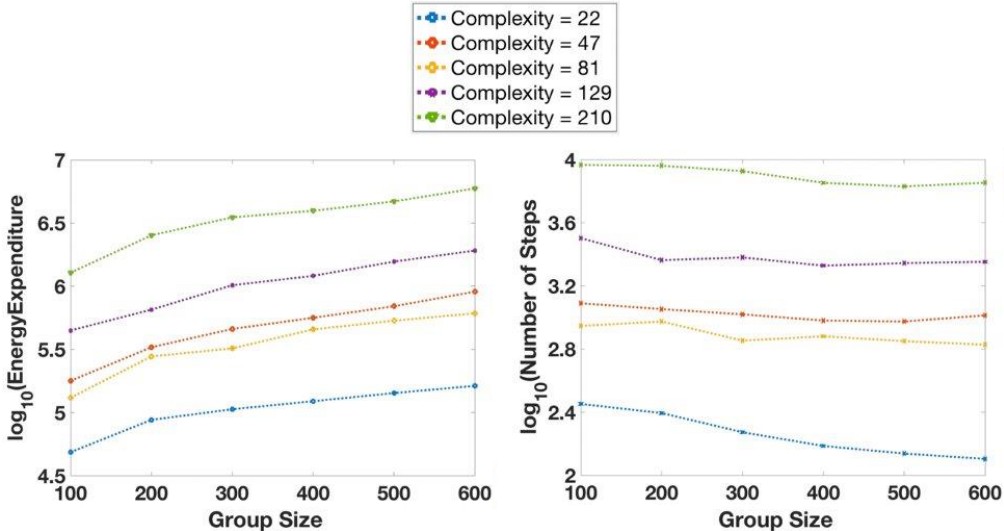

**Figure 12.** Number of steps to solve the maze and the associated energy expenditure as a function of search group size used to solve the maze, for 5 different maze complexities. The graphs on the leftplot the energy expenditure while the graphs on the rightplot the number of steps.

We note that the entries in Table 4, while commonsensical, are somewhat arbitrary, and it is relatively easy to imagine a practical situation where a different cost applies to specific direction choices. This, in turn, introduces some quantitative changes to the two sets of graphs shown in Figure 12.

**Table 4.** Cost estimates for all different possible directions/actions of movement.

| Direction | Cost |
|-----------|------|
| Hover | 0.5 |
| Forward | 1 |
| 90° turn | 1.5 |
| Backward | 2 |

### 4.4. Performance Measures

#### 4.4.1. Benchmark Comparison

As a benchmark, we compared the performance of our solution with another study that used ACO coupled with Fuzzy Logic, and also presented results in terms of the number of moves needed to solve the maze [89]. For the purpose of benchmark comparison, we set up mazes to the other study with the same sizes of 8 × 8 and 15 × 15 units. See Section 5.1.1 for the results.

#### 4.4.2. Cumulative Effort (Steps)

We equate the cumulative work of the swarm to the number of steps taken by them. Agents act once per iteration, but we only count this if it resulted in displacing the agent (i.e., we ignore the action of staying put). The summation of the steps taken by all agents in all iterations leading to the discovery of the target is the measure of the cumulative effort of the team of agents.

### 4.4.3. Estimated Energy Cost

Similar to the above, is the energy expenditure of the swarm; for example, due to the limitations in search agents' battery lifetime. The agents' battery powers both its mechanical and communication functions, which results in faster depletion of the portable energy source and makes the issue crucial in the success of the operation. As elaborated below, the energy expenditure of each possible change to the ongoing linear forward motion (which itself is represented as "one step forward") is accounted for through a modification of the forward motion by a multiplicative factor, presented in Table 4. The cost for moving backward is highest, as we envision the backward motion as a complete halt followed by a reversal in direction, which would require the maximum energy.

### 4.4.4. Assumptions and Considerations

Considering all agents expend comparable amounts of energy for communication with each other, we approximate it as a constant for a particular group size and neglect it in energy calculations to simplify simulations at this stage of analysis. The assumption is that knowing a group size allows designers to add the energy needed for the communication as an additive constant.

The main variable component in energy consumption between agents of a group is the directed motion of the search agents through their consecutive steps. Continuing in the same direction of motion is always cheaper than introducing a displacement in direction. As our agents are limited to a four-directional motion (in addition to an option to hover), the only direction changes possible are a 90°, to either side, or a 180° turn, meaning a backward motion. Considering the different amounts of energy needed to slow down (control speed) and change direction in each motion possible, a motion-based cost system was developed, as shown in Table 4.

## 5. Results and Discussion

### 5.1. Phase I: Maze Exploration

Although maze exploration was not a key motive for the research, it is of interest to discuss how well the heuristic part of the iAA system covers the maze during the search, as the mapping ability could lead to other applications in the search process.

As can be expected, the search results in almost a complete exploration (more than 80%) of the search space, in a special case depicted by Case 3 in Figure 13, when the target is located at the very end of the perfect maze. Exploration is much lower in cases where the target was located either close to the start or midway through the maze, but is still above 50%, which implies that the iAA technique could also be used for that purpose.

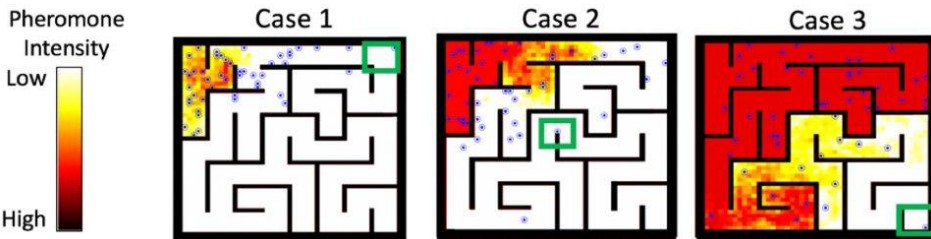

**Figure 13.** Three possible locations of a victim inside a maze and the resulting final pheromone levels. Figure 14 compares the resulting maze exploration and the number of steps taken to achieve it.

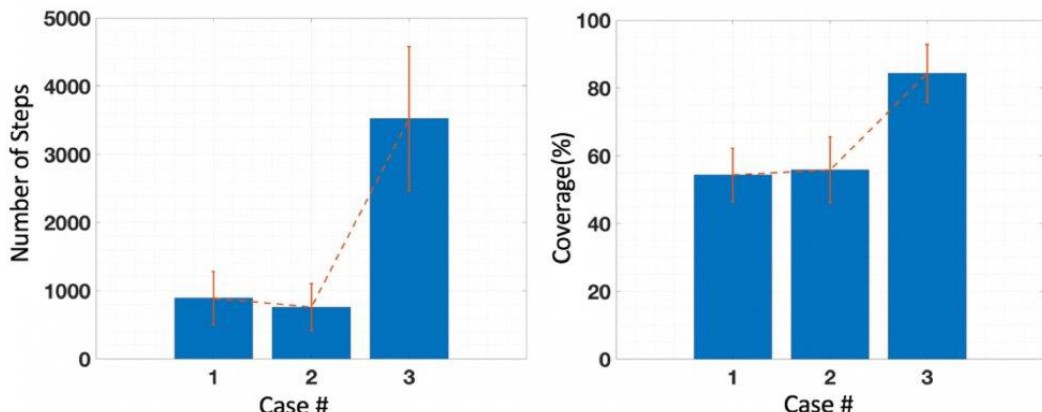

**Figure 14.** The achieved maze coverage (**right**), measured as the percentage of explored cells in the maze, and the corresponding number of steps (**left**) for each of the three mazes shown in Figure 13. Exploration is a by-product of the search process.

5.1.1. Benchmark Comparison

The performance of our iAA model was poorer than the benchmarked technique for both maze sizes when tested with search group sizes of two agents and five agents, which is a very small size for a swarm, as can be seen in Figure 15. We emphasize that we chose such small groups to make a comparison with the closest relevant reference. However, our iAA algorithm was quicker to converge and needed fewer steps, when the group size was increased to just 10 agents. A likely reason for the better performance in our system is due to the poor exploration capability of small group sizes, such as those with 2–5 agents.

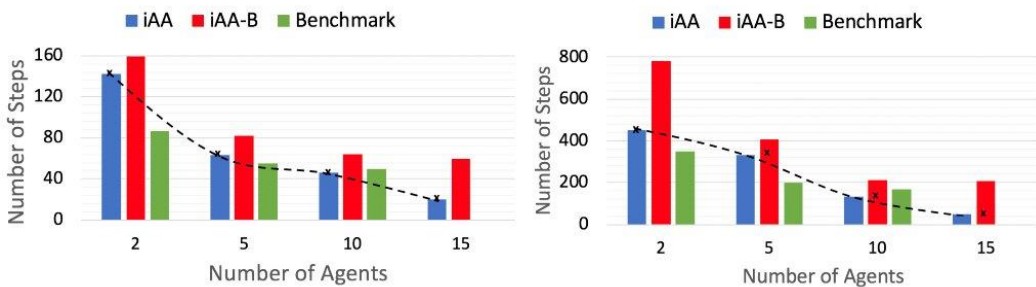

**Figure 15.** While the benchmark outperforms our approaches (iAA and iAA-B) for small group sizes, our technique iAA beats the benchmark [89] as the swarm size increases. Furthermore, an extended study shows that the trend of improvement in iAA is even more favorable as the swarm size continues to increase. This change in performance can be attributed to the swarming effect, which is better observed for a considerable search group size and does not show as much for a very small group (two agents or so).

For the maze sizes used, at least 7–10 agents would be necessary to promote quicker exploration that would lead to better results, as can be seen from the extrapolation of the trends in the results in Figure 15. These small group sizes were tested only for comparison with the results presented in the benchmark. Observing the interpolated line and looking at its trend as we change agent group size shows that our result is favorable compared to the benchmark, as can be seen from Figure 15. iAA also shows qualitatively similar behavior in a larger maze (right panel of Figure 15), with quantitatively better performance with 10 or more agents (quantitatively better than in the case of the smaller maze). Therefore, the more agents, and the bigger the maze, the more likely our proposed system is to outperform the benchmark study.

5.1.2. Cumulative Effort (Steps)

Figure 16 shows a comparison between the performances of the five models (when simulated with 100 ants each on the three mazes, with 30 repetitions each). The pure AA-based model did not introduce much of an improvement compared to a purely random solution. This lack of improvement can be attributed to the clustering effect of pheromone in pure AA, which is likely limiting the exploration of the maze. iAA is the best-performing algorithm among the five, closely followed by the performance of iAA-R.

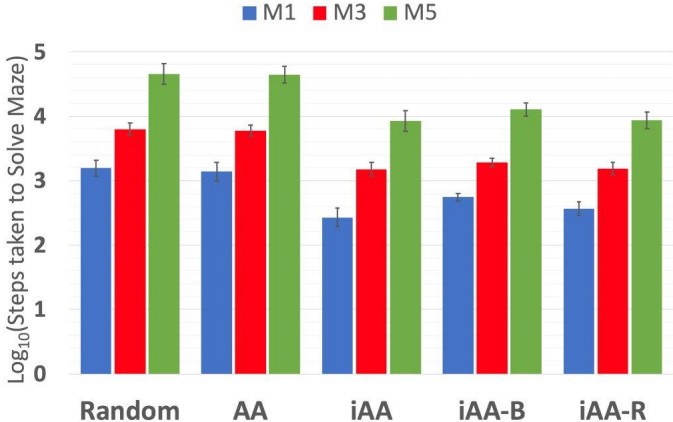

**Figure 16.** Comparing the performance of the 4 AA-based models and a random movement solution in solving the 3 of the sample mazes (M1, M3 and M5), which are of different sizes and complexities, with 100 ants each.

Contrary to expectations, iAA-B did not positively add to the performance of the iAA algorithm with its beacon initialization that was supposed to better guide the ants to the target. This is due to a trapping effect noticed in the simulation, as illustrated in Figure 17, where ants get trapped in nooks of the maze while being pulled towards the target.

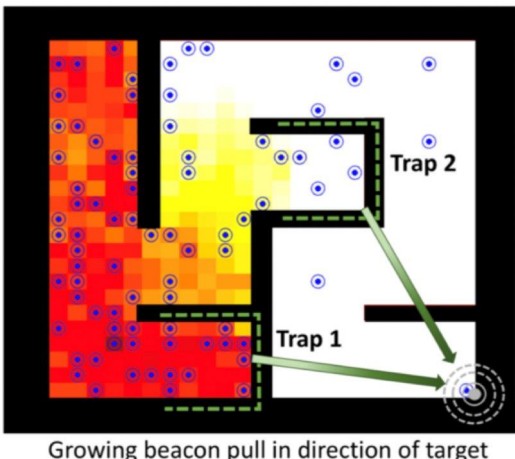

**Figure 17.** Inverted AA with beacon initialization for pheromone intensities can lead agents into the traps, marked as green-dashed U-shaped walls in the figure, as they tend to blindly follow a positive pheromone gradient and are unable to go around these obstacles.

As shown in Figure 18a, the growth in maze complexity is weakly supra-linear with the increase in maze size (cf. Table 3). The performance of the iAA strategy, measured by the number of necessary steps, in Figure 18b, with a fixed search group size, initially grows very fast with the increase in maze area, but appears to approach a saturation-like behavior above a certain maze size. Based on this and insights from related numerical experiments,

this point of near saturation determines the optimal maze size and complexity problem that can be solved by a search group of 100 agents.

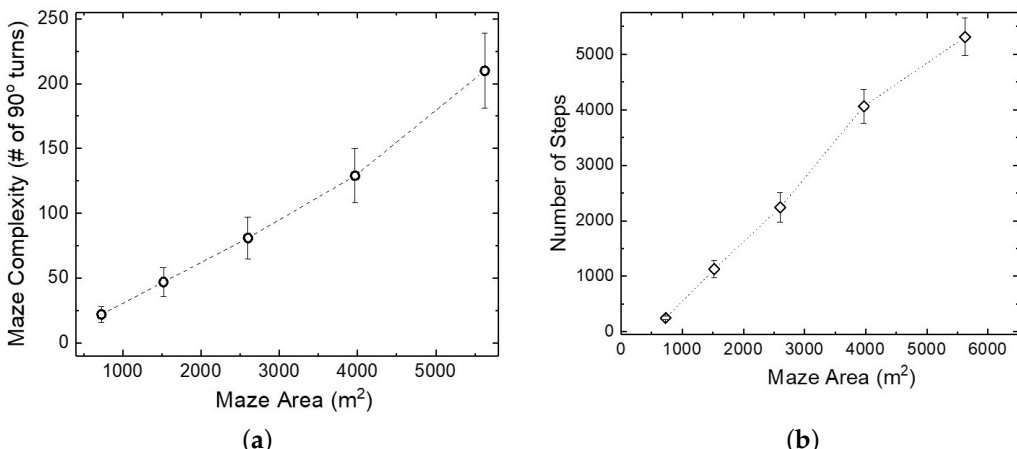

(**a**)　　　　　　　　　　　　　　　(**b**)

**Figure 18.** Performance of iAA algorithm as a function of variable maze size and maze complexity (cf. Table 3), with a constant search group of 100 agents. The plot in (**a**) shows the (weakly supra-linear) growth in complexity while (**b**) compares performance on the basis of the number of steps.

The weakly supra-linear dependence of the maze complexity on the maze size is to be expected, given that the number of "topological" choices (e.g., how to get around the maze), grows faster than the size itself. This can be checked through simple counting, by developing versions of Figure 11f while allowing the maze size to increase. The suggestion of the existence of the optimal size of the search agent group (Figure 18b) for a fixed maze size (while averaged overall a number of configurations at a fixed size), implies the presence of the non-linear dependence of the *Number of Steps* on the *Number of Agents*.

Based on our numerical experiments, it appears that the Search Task (Phase I) could have "too few" or "too many" agents, which is understandable in the context of the ant agent repellent action.

### 5.1.3. Estimated Energy Cost

### 5.2. Phase II: Signal Routing and Victim Extraction

To study the impact of changing the number of agents on the final solution in Dijkstra-based Phase II (the "rescue" part in SAR), the three performance metrics variables were generated—the total cost, the number of hops and the number of steps, in all cases as function of the number of agents. These metrics are shown in Figure 19. Here, number of steps implies the number of iterations of the algorithm execution to find a solution, while number of hops refers the number of logical hops in the relay network formed.

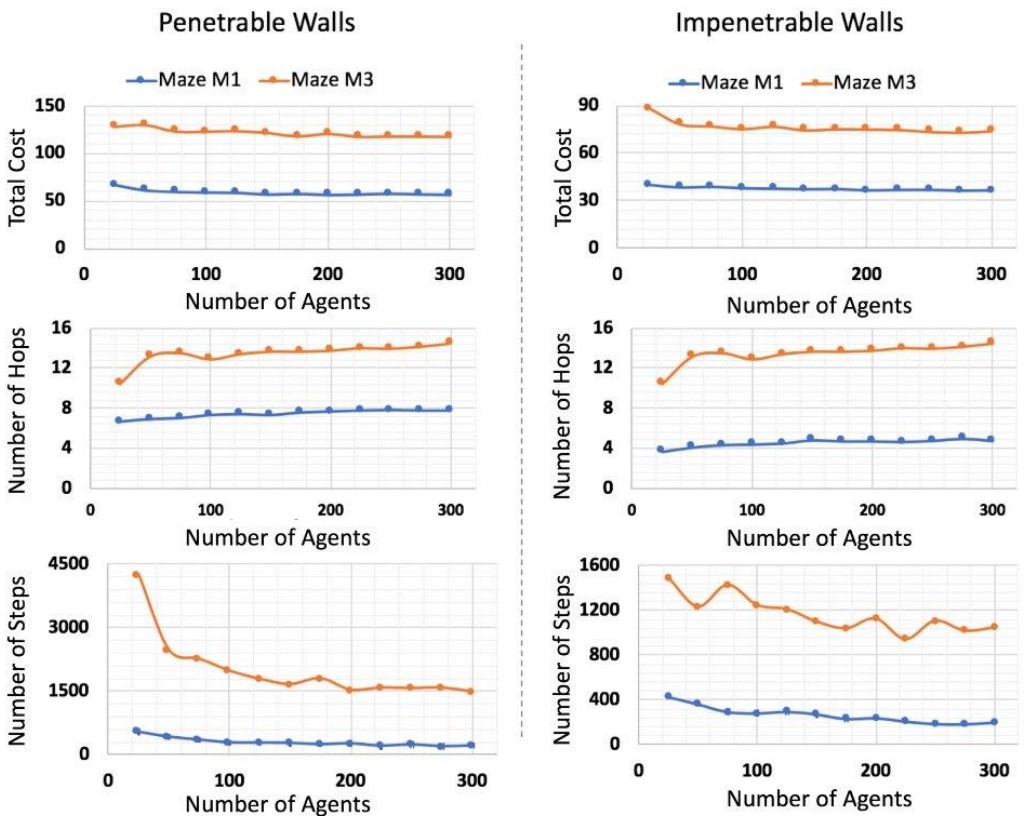

**Figure 19.** Performance analysis of Phase II in terms of number of steps (or iterations of the algorithm) taken to solve the maze, number of hops in the final relay network formed and the total cost of the relay path found (measured in distance), as tested for 2 maze complexities, M1 and M3.

For the steps count, one can intuitively assume that a small number of agents requires more steps to achieve the task, which slowly decreases as the number of agents increases and converges to a certain value. The reason is that fewer agents will need more time to form the shortest path, as the probability of it having more voids and going through the proposed search with the stopping algorithm is higher. The number of agents does not strongly affect the time taken to reach the target, as it depends on what path agents will follow. For the total cost, it decreases slightly when increasing the number of agents. By observing the simulation examples figure, one can deduce that the path looks almost the same; however, for the larger agents' population, the dependence is more smooth. Furthermore, the total cost when penetrable walls are considered is much less than for impenetrable walls. The number of steps is also lower for penetrable walls, but not by the same factor as the cost difference.

We have analyzed the performance for two different maze types, M1 and M2. As we have seen earlier in the paper, both size (area) and complexity of the maze play role in the performance (for Phase I). We can say, based on Figure 19, that both of these parameters are relevant in the performance of Dijkstra-based reverse routing in Phase II, as well. we can further say that the dependence is not one of simple linearity or even smoothness, as evidenced by less monotonous behavior of the Number of Steps and Number of Hops variables for maze M3 (it being more complex) than for the maze M1.

## 6. Conclusions and Future Work

To the best of our knowledge, combining variations of ACO/AA with Dijkstra to facilitate simultaneous localization and mapping (SLAM) in Search and Rescue scenarios has not previously been reported upon in the literature. We believe that the two-phase

approach, which was driven by the requirements of the real-world application, has the potential to be developed to a higher Technology Readiness Level in the near future.

### 6.1. Conclusions

SAR performed in two phases is summarized pictorially in Figure 20. The search starts with a victim being trapped in the maze-like environment, broadcasting an SOS signal. In Phase I, the search agents proactively explore the environment in search of the victim, using the iAA algorithm. Once the victim is spotted, the agents switch to the Phase II, and find the shortest routing path to communicate information about the victim's location to the base station, relying on specific implementation of Dijkstra algorithm and the results of the Phase-I search. Three different forms of the Ant Algorithm were explored and tested to devise the optimal search method. We studied the influence of the maze size and complexity, as well as the number of the search agents, together with the study of energy resources' dependence on problem and environment parameters. In ongoing efforts, we study the dynamic (time-varied) injection of the agents as a possible path towards additional performance improvement. We are in contact with agencies that may use this approach in their SAR training.

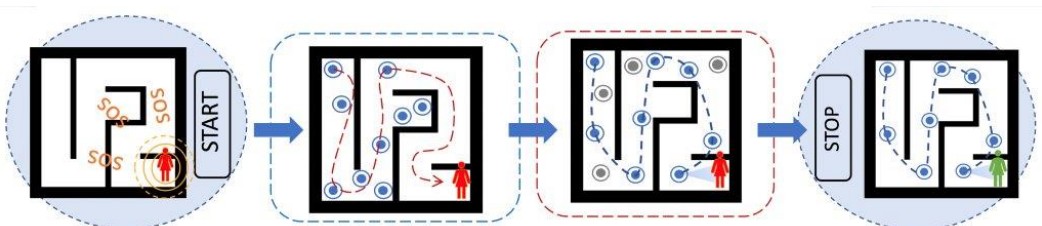

**Figure 20.** A summary view of the SAR process in two phases. The search starts with a victim being trapped in the maze-like environment, broadcasting an SOS signal. In Phase-I, the search agents explore the environment in search of the SOS signal, using iAA algorithm. After locating the source of SOS signal, the agents switch to Phase-II, and find the shortest routing path to communicate information about the victim's location to the base station. The remaining unused agents (marked in gray) can return to the base station.

### 6.2. Future Work

In Section 4.1.1, we stated that we simplified the computation and implementation of the system by discretizing a grid method similar to the one used in [72]. Naturally, future work would address the situations in which the number of agents (swarm size) and the size of the grid unit are not equal and provide problem specifications to the community that could be used as benchmark data sets for future work.

Furthermore, as stated in the introduction, the simulation is a 2D representation of the world. Extending this to a 3D model (both for the signal propagation through obstacles, as well as with the layout and structure of the building in mind), will be subject to an application-driven need to do so. The presented implementation and performance evaluation suffice to argue for the approach *in theory*; a real-world implementation will have to address a number of engineering issues, as well accounting for the 3D nature of the world. As we acknowledged before, there is a very active community working on all sorts of approaches related to, e.g., path finding for multiple devices [110], collision-free movement of a swarm [26], movement in formation [27] or through narrow spaces or bottlenecks [111]. Whether it be in 2D or 3D, we fully expect that any deployed solution using our approach will consist of a merger of multiple techniques, tailored to the specific problem at hand [112]. As we discuss in [15], applying nature-inspired solutions *for the sake of it*, while popular, is problematic. Future work will consider the practical aspects of implementing and deploying our solution for a swarm of UAVs, and will certainly reflect on the way in which the benefits of our approach can best be combined with both

nature-inspired [113] as well as other, non-nature-inspired approaches so as to optimize performance.

Finally, in future work, we would like to allocate some time to investigate the complexity of the mazes further. Preliminary investigations in which we compared optimal path lengths in randomly created mazes yielded results that informed the work presented in this article; however, the results themselves were omitted, mainly for the sake of brevity. A real-world study of mazes and operational environments for SAR swarms is required for work targeting the actual deployment of a drone swarm.

**Author Contributions:** Conceptualization and Methodology: D.R., A.F.I. and F.S.; Software and validation: Z.H., A.A.Z., D.R. and A.F.I.; Formal Analysis: Z.H., A.A.Z., D.R., F.S., H.H. and A.F.I.; Investigation: Z.H., A.A.Z., D.R., H.H. and A.F.I.; Data curation: Z.H., A.A.Z., D.R., F.S., H.H. and A.F.I.; Writing—original draft preparation: Z.H., A.A.Z. and A.F.I.; Writing—review and editing: Z.H., A.A.Z., D.R., F.S., H.H. and A.F.I.; Visualization: Z.H., A.A.Z., A.F.I. and H.H.; Supervision: A.F.I.; Project Administration: A.F.I. and D.R.; Funding acquisition: F.S., A.F.I. All authors have read and agreed to the published version of the manuscript.

**Funding:** We acknowledge the UAE 2013 ICTFund grant "Bio-inspired Self-organizing Networks".

**Data Availability Statement:** Data are available upon reasonable request from the corresponding author.

**Acknowledgments:** We thank Sami Muhaidat (KUST) for his advice about path loss models. We thank Patrick Grosfils (Universite Libre de Bruxelles) and R. Mizouni (KUST) on encouraging exchanges. A.F.I. thanks Cornell University for the hospitality during a part of this project.

**Conflicts of Interest:** The authors declare no conflict of interest.

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
