# Peer review of "Search and Rescue in a Maze-like Environment with Ant and Dijkstra Algorithms"

_drones, doi:10.3390/drones6100273_

Round 1

Reviewer 1 Report

Dear Authors,
In my opinion content of the article is consistent with subject matter of the journal.
However, there are numerous
issues that need to be clarified or corrected:

1. The 3d motion of UAVs is stated as extremely beneficial in the introduction, however all the design is presented in 2D.

2.  The paper  proposes SLAM algorithm based on ants algorithm. The mapping stops when path to goal is found.  No SLAM reference can be found in the references.

3. When the trajectory to the victim is found in the phase 1 by random exploration, it is stated to be near optimal in the abstract of the paper, However, it is  the first trajectory found, the computational time is the only criterion therefore it should not be called near optimal

4. The work needs benchmark comparison using relevant swarm size.

5. It is not clear what is the novelty of the proposed solution.

6. The simulation environment, paremeters and conditions are not clearly presented.

7. The choice of simulation parameters alpha and c in (1) is not explained, it seems to be just guess.

8. The algorithm does not take into account the dynamics of UAVs at all.

Author Response

We would like to thank Reviewer 1, as addressing comments and concerns expressed in their report helped us improve the manuscript. Below, Reviewer 1 comments are in bold italics, and our itemized answers and actions we undertook follow the Authors’ reply.

1.1 The 3d motion of UAVs is stated as extremely beneficial in the introduction, however, all the design is presented in 2D

Authors’ reply:

We thank Reviewer 1 for this comment and agree that clarification is necessary.

The introduction mentions that drones can operate in 3D space and emphasizes that this is indeed a significant benefit of UAVs. We mentioned early in Section 2.1 that our simulations are only in 2D space.

As we argue (in the updated manuscript), the environment can be seen as a stacked set of horizontal slices, one for each drone to operate in. This will allow us to avoid the drones operating in the same space (collision).

Furthermore, in the simulation, we explore the ability of the swarm to locate victims swiftly. The simulated environment, which is, e.g., representative of a single floor in a multi-story building, does not contain obstacles that require changes in the z-axis to circumnavigate. Therefore, there is relatively little benefit to extending the simulation to full 3D representation.

Finally, we would like to comment that we expect that TRUE 3D swarm exploration would (a) be useful in, e.g., underground caverns and would (b) require significantly more drones. However, the algorithms would not change; therefore, the simplified simulations used suffice to make a point of the paper.

1.2 The paper proposes SLAM algorithm based on the ants algorithm. The mapping stops when the path to the goal is found.  No SLAM reference can be found in the references.

Authors’ reply

We thank Reviewer 1 for this comment and agree that this should have been addressed. We have done so by specifically mentioning SLAM in the introduction and discussing (briefly) that the use of drones was originally primarily driven by the military application, of which the mapping of environments was a major aspect.

The following references have been added:

[6] A Gaussian Process model for UAV localization using millimeter wave radar

[7] Exploration-Based SLAM (e-SLAM) for the Indoor Mobile Robot Using Lidar

[8] Simultaneous Localization and Mapping (SLAM) and Data Fusion in Unmanned Aerial Vehicles: Recent Advances and Challenges

[16] CNN-Based Dense Monocular Visual SLAM for Real-Time UAV Exploration in Emergency Conditions

[23] Autonomous Exploration of Unknown Indoor Environments for High-Quality Mapping Using Feature-Based RGB-D SLAM

1.3 When the trajectory to the victim is found in phase 1 by random exploration, it is stated to be near-optimal in the abstract of the paper; however, it is the first trajectory found, and the computational time is the only criterion; therefore, it should not be called “near optimal.”

Authors’ reply:

We agree with the reviewer that such phrasing was in error, so we have modified the said phrase. This being said and done, we feel it is important to point out that the search conducted as described in our manuscript is not purely random, as there is an effective bias due to the presence of (presumably weak but detectable) signal from the victim (such as an infrared signature, or a call from personal electronics, or wearable electronics).

1.4 The work needs benchmark comparison using relevant swarm size.

Authors’ reply

We thank Reviewer 1 for bringing this up, and we partially agree that better benchmarking could have been done. In principle, one can always construct an increasing series of benchmark tests that help “converge” towards an ideal check. We consider such a task complex enough that it should be a separate report. Instead, we have identified one example where there are considerable initial similarities but nuances of our method point to quantifiably improved performance even for a relatively small swarm size, with a clear trend that the increase in the swarm size favors the version by our approach. We think this satisfies “a benchmark” criterion, while we agree that “the benchmark” might be missing from this manuscript.

We emphasize this in a (short added) statement in Section 3.1 and Section 6.2.

We furthermore elaborate on the benchmarking in Section 4.4.1.

1.5 It is not clear what is the novelty of the proposed solution.

Authors’ reply

To the best of our knowledge, this combined method (variations in Ant Algorithm for search and Dijkstra for rescue) has not been published before, which defines the novelty. We have stated this in the (new) opening paragraph of the Conclusion.

1.6 The simulation environment, parameters, and conditions are not clearly presented.

Authors’ reply

We thank Reviewer 1 and agree that this needed to be addressed. We have added subsection 4.2.1 to discuss the hardware and software used for the data collection. Furthermore, 4.1.2 provides the values for two constants (wall attenuation factor; frequency channel). In our view, the formula provided for Equation 5 is appropriate to calculate realistic values for our simulation. See also the answer to your following comment.

1.7 The choice of simulation parameters alpha and c in (1) is not explained; it seems to be just guess.

Authors’ reply

We thank Reviewer 1 for bringing this up, as we should have been more specific in the initially submitted manuscript. We have added an explanation for how we proceeded at the end of Section 2. Given the specific nature of the application we discuss in this manuscript, we have explored the ranges of values for both parameters, namely a in the range [1.8, 2.2] and c in the range [18.0, 21.0]. We find no reason to step away from the ACO/AA “dogma” on this issue.

1.8 The algorithm does not take into account the dynamics of UAVs at all.

Authors’ reply

We thank Reviewer 1 for this comment. We have added the following clarification in the text “At the present stage of the development and application of this hybrid model, we only account for a controlled discretized motion of UAVs, as specified, for example, in Table 4”. This can be found in Section 3.1.4, which has been added in its entirety to address the reviewer’s concerns.

Reviewer 2 Report

The paper covers an interesting topic - use of drones in SAR operations with an added bonus of using drones as an ad hoc network. 

  1. The main issue with the draft is that its structure obscures both the topic and the contribution of the paper. As it seems, the main focus of the technical part of the paper is the proposed method for terminating exploration in a multi-agent system and forming a connection from the target to the starting point. This means that the subject of the study is 2D maze-like environment + search algorithm with multiple agents + the question of forming a connected route. The SAR has only application-level relevance, and it is actually unclear if/when the proposed method would be relevant for SAR operations (mazes are not the most typical disaster site, if a victim is located it might be easier to produce an alarm signal instead of forming some kind of ad hoc network - it is all a matter of practical needs, first aid, etc., which is well outside the scope of this paper) but it nonetheless clear that the theoretical problem solved in the paper in important. So, the paper can mention SARs and drones as motivation for the study, there is no need to discuss them at length. Indeed, now references [1-16, 18] and others are cited in discussion of UAVs, which are only tangentially relevant to the topic of the paper. Similarly, the discussion of ad hoc networks should be limited to the amount needed to understand the contribution of the paper. Reducing discussions of drones, ad hoc networks, SAR operation and instead making a strong and clear statement regarding what the paper contributes (in clear technical language) would make it easier to read and improve its usefulness to the reader. Indeed the discussion of drones, ad hoc networks, and SAR operations should serve as an illustration why the posed problem has practical significance and point to existing solutions to the most similar problems already tackled in the field. 

  1. This leads to the second topic. As the main contribution of the paper seems to be the modification of a graph search method, it would be beneficial to have a separate section describing 1) how the problem that the paper tries to solve is different from the most similar problem that already has a known solution, 2) upsides and downsides of the existing solutions that could apply to the problem the paper tries to solve, if there are any. If the proposed solution is the first to solve this class of problems, it would be very important to note this in the abstract and the introduction.

Some other remarks. 

  1. State of the art review (section “Background” in the paper) can be improved. Currently, the structure of the section is not straightforward, for example subsection “2.1. Indoor Search and Rescue (SAR) operations for swarms of drones” seems to be a double of subsection “2.2.4. UAVs in Search and Rescue operations”.

  1. English and grammar can be improved:

page 1: “Nowadays UAVs are technological advanced devices” - should be ‘technologically’

page 1: “depending on their intended use the required specifications” - missed ‘and’ after ‘use’.

Author Response

We would like to thank Reviewer 2, as addressing comments and concerns expressed in their report helped us improve the manuscript. Below, Reviewer 2 comments are in bold italics, and our itemized answers and actions we undertook follow the Authors’ reply.

2.1 The main issue with the draft is that its structure obscures both the topic and the contribution of the paper. As it seems, the main focus of the technical part of the paper is the proposed method for terminating exploration in a multi-agent system and forming a connection from the target to the starting point. This means that the subject of the study is 2D maze-like environment + a search algorithm with multiple agents + the question of forming a connected route. The SAR has only application-level relevance, and it is actually unclear if/when the proposed method would be relevant for SAR operations (mazes are not the most typical disaster site, if a victim is located it might be easier to produce an alarm signal instead of forming some kind of ad hoc network - it is all a matter of practical needs, first aid, etc., which is well outside the scope of this paper) but it nonetheless clear that the theoretical problem solved in the paper in important. So, the paper can mention SARs and drones as motivation for the study, there is no need to discuss them at length. Indeed, now references [1-16, 18] and others are cited in discussion of UAVs, which are only tangentially relevant to the topic of the paper. Similarly, the discussion of ad hoc networks should be limited to the amount needed to understand the contribution of the paper. Reducing discussions of drones, ad hoc networks, SAR operation and instead making a strong and clear statement regarding what the paper contributes (in clear technical language) would make it easier to read and improve its usefulness to the reader. Indeed the discussion of drones, ad hoc networks, and SAR operations should serve as an illustration why the posed problem has practical significance and point to existing solutions to the most similar problems already tackled in the field.

Authors’ reply

We thank Reviewer 2 for opening up the discussion on several issues, which we answer below:

This is indeed a complex, technical paper, as we aimed to establish what is doable within certain simplifying assumptions but still application-worthy in the context of SAR operation.

We contend that mazes are a good representation of disaster-struck SAR areas, for example, as representations of individual floors of buildings. Granted, in a disaster-type situation, one wouldn’t expect all the walls to retain the pre-disaster geometry or integrity, but implementing that level of reality is worthy of a separate project. Furthermore, we make no assumption that the maze generated for the purpose of performance investigation reproduces a pre-disaster “floor plan”. It is used as a convenient abstraction for a complex environment to navigate, of which the drones have no prior knowledge. As such, the geometry produced could just as well be that of a collapsed building (with the one restriction that there are no unreachable areas).

We agree that this is primarily an application-oriented manuscript. This being said, it shouldn’t invalidate it from being published. The manuscript contains the following:

(a)        the output of simulation that motivates additional biased random walk on a discrete maze (biased by the presence of a signal, so it is, in a way, quasi-random), and

(b)       an example of how an outcome of Ant Algorithm search could be fed into the Dijkstra Algorithm

2.2 This leads to the second topic. As the main contribution of the paper seems to be the modification of a graph search method, it would be beneficial to have a separate section describing 1) how the problem that the paper tries to solve is different from the most similar problem that already has a known solution, 2) upsides and downsides of the existing solutions that could apply to the problem the paper tries to solve, if there are any. If the proposed solution is the first to solve this class of problems, it would be very important to note this in the abstract and the introduction.

Authors’ reply

We thank Reviewer 2 for the suggestion. To the best of our knowledge, the setup as we suggest: (a) varied maze configurations, (b) stationary victim emitting either biological signature signal (body heat) or wearable electronics signal, (c) versions of AA + Dijkstra explored, is unique enough that we haven’t found a proper comparison, despite a reasonably thorough search.

To be clear, we are not claiming that the proposed hybrid approach is the best solution since we haven’t conducted a formal analysis to that effect, but based on the intuition from trying to solve this with PSO (and PSO hybrids), we would be surprised if there is an effective way of solving this problem so much more efficiently that would cast aside any need for the use of the solution we propose.

2.3 State of the art review (section “Background” in the paper) can be improved. Currently, the structure of the section is not straightforward, for example subsection “2.1. Indoor Search and Rescue (SAR) operations for swarms of drones” seems to be a double of subsection “2.2.4. UAVs in Search and Rescue operations”.

Authors’ reply

We thank Reviewer 2 and agree that there is a potential for confusion in how some parts of our “background” are organized. To address this concern, we have re-ordered some text and streamlined the sub-titles. The changes are marked in red (in the resubmitted and revised document)

2.4 English and grammar can be improved:

page 1: “Nowadays UAVs are technologically advanced devices” - should be ‘technologically’

page 1: “depending on their intended use the required specifications” - missed ‘and’ after ‘use’.

Authors’ reply

We thank Reviewer 2, and we wholeheartedly agree. We have worked on correcting quite a few typos and grammar errors.

Reviewer 3 Report

In this paper, Search and Rescue effort in the event of a disaster where timely discovery and help are very important is investigated. Ant Algorithm (AA) is adapted for the phase of Search, and Dijkstra’s algorithm is then used in the Rescue phase.

Since Ant algorithm is not new, and is designed to be applied in such scenarios as those from this paper, it is not clear for me which the novelty of this work is. Moreover, Dijkstra’s algorithm used in the rescue phase is also common sense since a graph can be constructed with the positions of the robots and their connections (arcs). Passing over this main concerns of validation of known ideas, comparisons should be made between proposed method and other methods, e.g., from Table 1.

Another concern is about communication. The victim (that has to be saved) has the possibility to emit SOS signals, and the communication between robots is limited. It is odd.

Some literature review about Ad Hoc networks, DTN networks, disrupted communication, disaster areas, and algorithms/protocols used in these scenarios should be conducted. For instance, the following two papers could be a good start:

Nazib, R.A.; Moh, S. Routing Protocols for Unmanned Aerial Vehicle-Aided Vehicular Ad Hoc Networks: A Survey. IEEE Access 2020, 8, 77535–77560 

Deaconu, A.M.; Udroiu, R.; Nanau, C.-Åž. Algorithms for Delivery of Data by Drones in an Isolated Area Divided into Squares. Sensors 2021, 21, 5472. https://doi.org/10.3390/s21165472

Author Response

We would like to thank Reviewer 3, as addressing comments and concerns expressed in their report helped us improve the manuscript. Below, Reviewer 3 comments are in bold italics, and our itemized answers and actions we undertook follow the Authors’ reply.

3.1 In this paper, Search and Rescue effort in the event of a disaster where timely discovery and help are very important is investigated. Ant Algorithm (AA) is adapted for the phase of Search, and Dijkstra’s algorithm is then used in the Rescue phase.

Since Ant algorithm is not new, and is designed to be applied in such scenarios as those from this paper, it is not clear for me which the novelty of this work is. Moreover, Dijkstra’s algorithm used in the rescue phase is also common sense since a graph can be constructed with the positions of the robots and their connections (arcs). Passing over this main concerns of validation of known ideas, comparisons should be made between proposed method and other methods, e.g., from Table 1.

Authors’ reply

We appreciate the concern raised by Reviewer 3. We agree that these two methodologies might be a common sense approach, but to the best of our knowledge:

(1)        they have not been used in a hybrid combination the way we proposed and developed here.

(2)        Additional aspect of novelty are the details of the environment – the variations in maze parameters, as well as the signal-biasing of the search (some approaches do use maze, but do not include the signal bias emanating from either biological (heat) signature of the hypothesized victim, or from their wearable electronics.

Lastly, the maze parameters are explored at some length.

3.2 Another concern is about communication. The victim (that has to be saved) has the possibility to emit SOS signals, and the communication between robots is limited. It is odd.

Authors’ reply

We thank Reviewer 3 for raising these concerns. We have now made clarifications to indicate that different passive signals emitted without the victim’s effort can be used.

It is also possible that our choice of words for communication between UAVs/robots is unfortunate. The idea was to emphasize that simple communication is sufficient to “solve” this problem.

3.3 Some literature review about Ad Hoc networks, DTN networks, disrupted communication, disaster areas, and algorithms/protocols used in these scenarios should be conducted. For instance, the following two papers could be a good start:

Nazib, R.A.; Moh, S. Routing Protocols for Unmanned Aerial Vehicle-Aided Vehicular Ad Hoc Networks: A Survey. IEEE Access 2020, 8, 77535–77560

Deaconu, A.M.; Udroiu, R.; Nanau, C.-Åž. Algorithms for Delivery of Data by Drones in an Isolated Area Divided into Squares. Sensors 2021, 21, 5472. https://doi.org/10.3390/s21165472

Authors’ reply

We thank Reviewer 3 for the suggestions. These are now references 20 and 21 in the updated and re-submitted manuscript.

Additionally, references 6,7, 8, 16, and 23 were added to improve the literature list for UAVs, networks, Localization, SLAM, and drone-to-drone communication, in the context of disaster relief efforts.

Round 2

Reviewer 2 Report

The reviewer thanks the authors for the replies, improvements to the draft and clarifications in the comments.

Reviewer 3 Report

The authors addressed my concerns.